# Country-Based COVID-19 DNA Sequence Classification in Relation with International Travel Policy

Elis Khatizah [1,*] and Hyun-Seok Park [1,2,*]

1  Department of Computer Science and Engineering, Ewha Womans University, Seoul 03760, Republic of Korea
2  Interdisciplinary Program, Department of Computational Medicine, Ewha Womans University, Seoul 03760, Republic of Korea
*  Correspondence: elis_khatizah@ewhain.net (E.K.); neo@ewha.ac.kr (H.-S.P.); Tel.: +82-1067050128 (E.K.); +82-232773513 (H.-S.P.)

**Abstract:** As viruses evolve rapidly, variations in their DNA may arise due to environmental factors. This study examines the classification of COVID-19 DNA sequences based on their country of origin and analyzes their primary correlation with the country's international travel policy. Focusing on DNA sequences from nine ASEAN countries, we conducted a two-class classification to distinguish sequences from individual countries and mixed sequences from others. The sequences were initially dissected into 200 base pair units, and a deep-learning method was employed to construct a model. Our results showcase the capacity to differentiate DNA sequences with varying accuracy for each country. Additionally, the index of international travel policy, which reflects how countries implemented varying levels of restrictions regarding inbound travel, several months before the sequence collection date, moderately correlated with the classification accuracy within each country. This finding suggests a preliminary insight that a country's pandemic management might influence the variation in the DNA virus, determining whether these sequences will evolve distinctly from those of other countries or exhibit similarities.

**Keywords:** ASEAN; COVID-19; DNA sequences classification; international travel policy; pandemic management

## 1. Introduction

Over three years, the COVID-19 global crisis has offered valuable lessons that can benefit humanity when facing similar disasters in the future [1]. The outbreak also indirectly highlights the varied responses and approaches adopted by different countries in managing the disease. Resulting proposed actions, such as the implementation of strict measures and financial support, notably proved effective in reducing infection and fatality rates [2]. Furthermore, a substantial amount of genomic data related to SARS-CoV-2, the virus causing COVID-19, has been systematically stored in popular databases. This wealth of information allowed scientists to extract insights and knowledge for effectively combating the pandemic. Utilizing this dataset, numerous studies have been conducted such as differentiating the DNA sequence of the COVID-19 virus from other diseases with similar symptoms [3–5] and classifying these sequences based on virus variants [6,7]. By leveraging AI technology, these efforts can provide benefits for cost-effective disease detection.

Initially, no significant mutations were observed in the COVID-19 virus during the first 8 months following the World Health Organization's declaration of it as a global threat [8]. However, a series of variants of concern gained prominence shortly thereafter, starting with the Alpha variant that emerged in September 2020 [9] and concluding with the Omicron variant, which became the predominant variant worldwide as of January 2022 [10]. Studies indicate that human migration (gene flow) plays a crucial role in virus evolution [11], allowing for the generation of many variations. The increased occurrence of distinct mutations in European, Asian, and North American sequences suggests the

virus's tendency to cluster geographically [12]. Moreover, analyzing the variations in the coronavirus genome among 10 countries, including the Czech Republic, France, Thailand, the USA, Japan, Taiwan, China, Australia, Greece, and India, concludes that the structure of the coronavirus genome significantly differs among countries [13]. In a specific area, a genome-wide analysis of circulating SARS-CoV-2 strains has been conducted to identify the emergence of novel co-existing mutations and trace their geographical distribution within India [14]. Thus, the genetic variances observed among the strains from diverse locations can be linked to their geographical distributions.

When considering the impact of the COVID-19 outbreak, a comprehensive analysis of the SARS-CoV-2 genome sequences isolated from individuals in six geographic areas reveals correlations with fatality rates in various countries [15]. This correlation is also supported by studies indicating a positive connection between the occurrence of specific gateway mutations and mortality [16,17]. Conversely, some studies suggest no association between SARS-CoV-2 variants and mortality rates [18–20]. Nevertheless, the diverse structure of the coronavirus genome among countries is a critical aspect that requires investigation for vaccine development [13]. In addition to extensive work in phylogenetic-based approaches, which includes a specific profiling method for DNA sequences aimed at the accurate analysis of SARS-CoV-2 genomes at the genus or species levels [21], it is equally important to explore country-based classifications of sequences. To the best of our knowledge, no existing literature has explicitly conducted a classification of DNA sequences based on the country where they come from. While we cannot ignore the fact that each country has varying land areas that may influence the level of virus variation in each region, recognizing the distinction of DNA sequences, a pivotal component in the genomic data, based on the country of origin could potentially deepen our understanding of how the virus evolves within the area.

As the home to more than 9% of the world's population and a region with a surge in infection cases since March 2020 [22], it is crucial to investigate the role of geographical proximity in shaping the genetic structure of the SARS-CoV-2 genome in the Association of Southeast Asian Nations (ASEAN) countries. Several strains that were isolated before the national implementation of border control exhibited a high degree of invariance, while others demonstrated approximately 80% of synonymous mutations, suggesting possible ongoing virus adaptation in the region [23]. Focused on DNA, research related to sequence profiling has been carried out regionally, involving five ASEAN countries with a relatively limited number of sequences [24]. To enhance the analysis with a larger dataset and provide clear information on how the virus' DNA differs between countries, particularly where the countries are geographically close to each other, this paper purposes to classify SARS-CoV-2 DNA sequences based on the country using a deep learning model. Concentrating on nine ASEAN countries, we also primarily explore the connection between the classification results and each country's international policy restrictions to gain insight into how the virus behaves in different places under different rules.

## 2. Data and Method

### 2.1. Data

There are three datasets utilized in this study: DNA sequences, the number of confirmed cases and deaths from COVID-19, and the international travel policy data. The first and primary data were sourced from the official Global Initiative on Sharing All Influenza Data (GISAID) website [25]. Our focus centered on SARS-CoV-2 nucleotide sequences categorized as complete (sequence length > 29,000 bp) and high coverage (only entries with less than 1% undefined bases) from nine ASEAN countries: Brunei, Cambodia, Indonesia, Laos, Malaysia, Philippines, Singapore, Thailand, and Vietnam. Specifically, we selected DNA sequences with a collection date range from 1 July 2021 to 31 December 2021. Referring to the website, in comparison with the other half-year periods over three years, each country submitted a relatively larger number of sequences during this timeframe,

as indicated in Table 1. This sufficient dataset is anticipated to provide comprehensive information, addressing the potential limitations associated with data scarcity.

**Table 1.** Number of SARS-CoV-2 DNA sequences submitted to the GISAID database by each ASEAN country in half-year periods over 3 years, categorized as complete and high coverage.

| Country Name | Number of Sequences Submitted in: | | | | | |
|---|---|---|---|---|---|---|
| | January–June 2020 | July–December 2020 | January–June 2021 | July–December 2021 | January–June 2022 | July–December 2022 |
| Brunei | 5 | 2 | 2 | 488 | 118 | 54 |
| Cambodia | 20 | 26 | 460 | 1487 | 578 | 86 |
| Indonesia | 204 | 581 | 4186 | 4428 | 715 | 332 |
| Laos | - | - | - | 862 | 233 | 12 |
| Malaysia | 165 | 326 | 1341 | 5777 | 2800 | 848 |
| Philippines | 29 | 337 | 4293 | 3292 | 24 | 148 |
| Singapore | 1092 | 363 | 1664 | 7171 | 969 | 303 |
| Thailand | 307 | 144 | 2161 | 6532 | 1277 | 398 |
| Vietnam | 121 | 55 | 275 | 1658 | 411 | 15 |

Additional information regarding the dataset downloaded from the GISAID website in the second half of 2021, taking into account variants of concern, reveals that the Delta variant dominates the variant distribution in ASEAN countries. As illustrated in Table 2, except for Cambodia and the Philippines, other countries possess DNA sequences comprising more than 94% Delta variant, with Brunei and Vietnam exclusively consisting of this variant in their datasets. In the subsequent classification process, we conducted the classification using the all available DNA sequences and also exclusively using the Delta variant sequences, aiming to observe the impact of this variant on the classification results.

**Table 2.** Variant distribution of SARS-CoV-2 DNA sequence contributed by ASEAN countries in GISAID during the second half of 2021.

| Country Name | The Distribution Percentage of Variant: | | | | | |
|---|---|---|---|---|---|---|
| | Delta | Omicron | Alpha | Beta | Gamma | Others |
| Brunei | 100.00% | | | | | |
| Cambodia | 72.02% | | 27.98% | | | |
| Indonesia | 97.70% | 0.07% | 0.02% | | | 2.21% |
| Laos | 99.65% | | 0.23% | | 0.12% | |
| Malaysia | 98.67% | 0.09% | | 0.31% | | 0.93% |
| Philippines | 70.47% | | 13.61% | 12.52% | | 3.40% |
| Singapore | 99.78% | 0.13% | 0.03% | 0.01% | | 0.06% |
| Thailand | 94.32% | 0.05% | 5.10% | 0.38% | | 0.15% |
| Vietnam | 100.00% | | | | | |

The second dataset consists of the impact of COVID-19 in each ASEAN country accessed from official data collated by Our World in Data [26]. This dataset includes the number of daily new confirmed cases, daily new confirmed deaths, cumulative confirmed cases, and cumulative confirmed deaths per million people. We focus on data spanning the past three years, from 1 March 2020 to 28 February 2023. By aggregating the daily value per week, we plotted the number of weekly confirmed cases and deaths, as depicted in Figure 1. Based on the figure, the pandemic conditions in most ASEAN countries were relatively severe from 1 July 2021 to 31 December 2021 as illustrated by the highest number of weekly confirmed deaths and a significant number of weekly confirmed cases per million people in this period. This contextual information is invaluable for analyzing the DNA sequences classification results in connection to the pandemic situation.

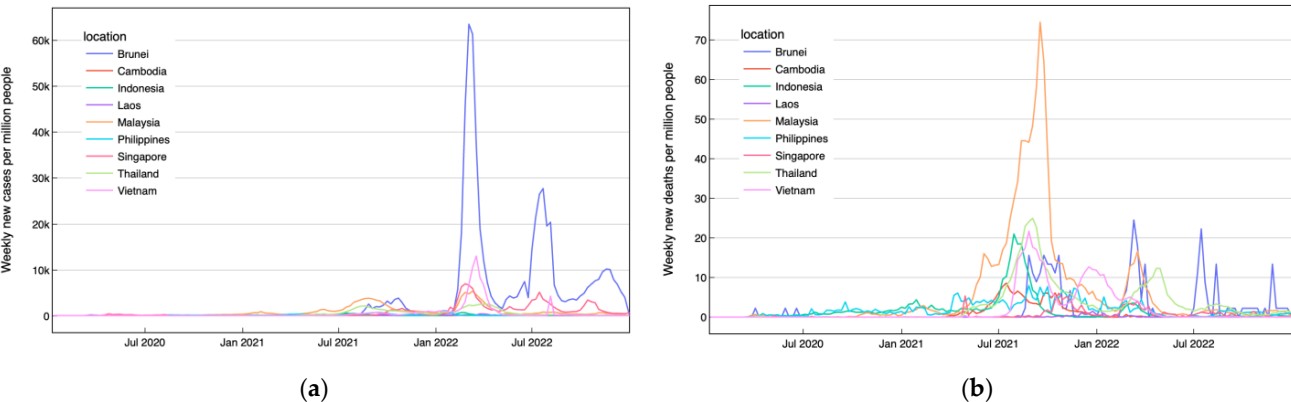

**Figure 1.** (**a**) Weekly confirmed cases per million people and (**b**) weekly confirmed deaths per million people in ASEAN countries over three years.

The third dataset consists of data accessed from The Oxford COVID-19 Government Response Tracker (OxCGRT) project. This comprehensive initiative has collected information on policy measures adopted to combat COVID-19 throughout the years 2020, 2021, and 2022 [27]. We downloaded an Excel file containing countries from around the world and their corresponding restriction indicators, as shown in Figure 2. Subsequently, we extracted data covering only the nine ASEAN countries for the year 2021. Given that the study focuses on classifying DNA sequences between countries, our primary concern was restrictions related to policies impacting individual movement between countries, specifically restrictions on international travel (c8), or what we later refer to as international travel policy. This indicator is used to record policies related to incoming foreign travelers to a certain country. It does not report restrictions on outbound travel and does not count citizen repatriation as a case of open borders (if all other inbound travel is restricted).

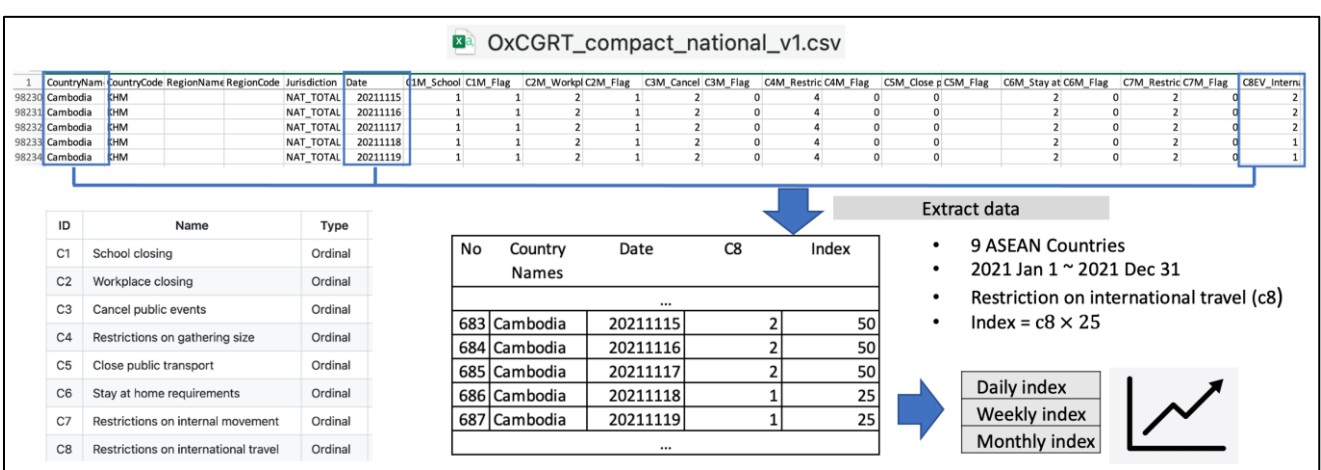

**Figure 2.** Processing international travel policy data into daily, weekly, and monthly index.

The level score of the international travel policy is recorded daily, indicating 'no restrictions' (scored 0), 'screening arrivals' (scored 1), 'quarantining arrivals from some or all regions' (scored 2), 'banning arrivals from some regions' (scored 3), and 'complete ban on all regions or total border closure' (scored 4). For ease of interpretation, the score is multiplied by a scalar of 25 to transform it into an index ranging from 0 to 100. Consequently, we obtain a daily index of the international travel policy, appearing in the last column of every row of the dataset. Next, by averaging the index per week or month, we can derive weekly and monthly indices, respectively. This process allows us to create graphical representations of the weekly or monthly international travel policy index for each ASEAN country throughout 2021.

### 2.2. Method

Figure 3 demonstrates the workflow of deep learning models used for classifying COVID-19 DNA sequences based on their country of origin. During the data processing phase, the sequences were initially downloaded and stored in nine separate FASTA files, corresponding to the countries they originate from. We conducted a two-class classification to distinguish sequences from a specific country from those of others. This is referred to as 'one-vs-all', where 1000 sequences from the country of interest were randomly selected versus 1000 mixed sequences from other countries (each contributing 125 sequences). For the country of interest that has a number of samples less than 1000, we included all available sequences in our analysis. These two groups of sequences were respectively saved in two separate TXT files. Subsequently, the sequences in each group were dissected into 200-bp units, labeled accordingly (1: unit from the country of interest, 2: unit from mixed countries), and integrated into a single file.

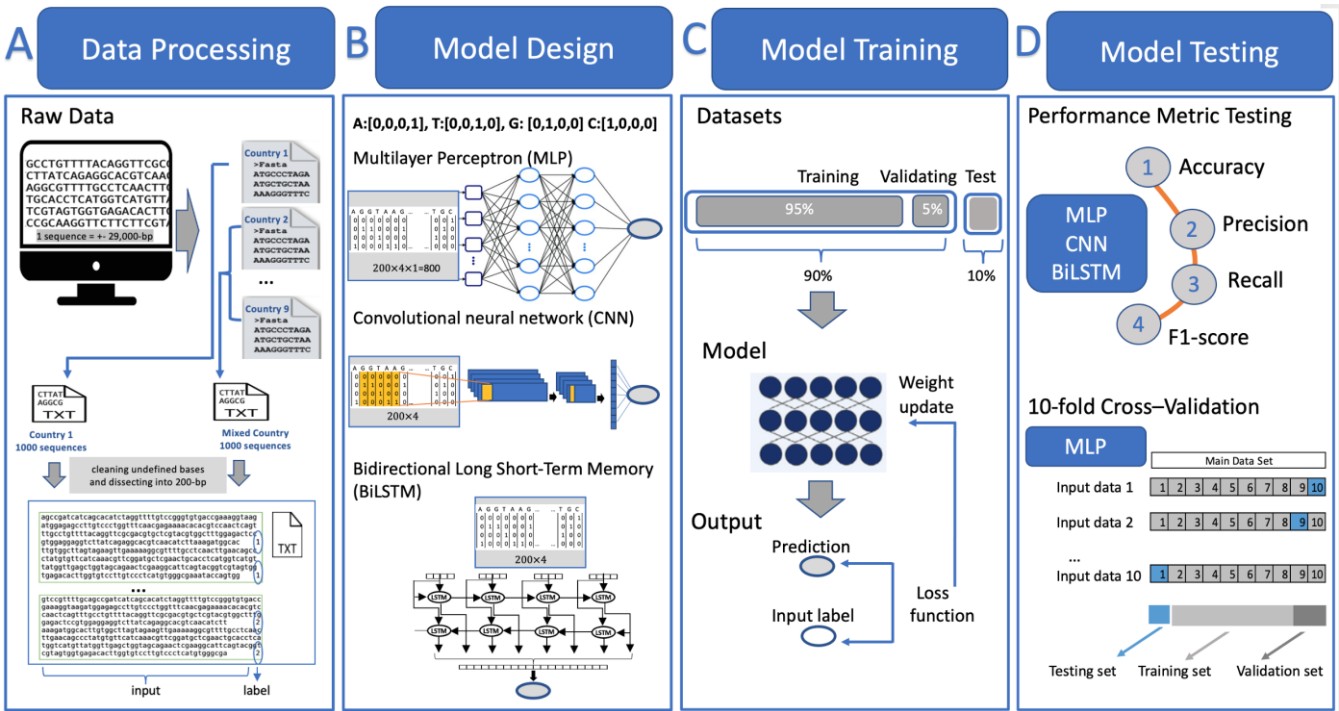

**Figure 3.** A diagram illustrating the workflow of deep learning models, including data processing, model design, model training, and model testing.

In the model design phase, we trained three models: Multi-Layer Perceptron (MLP), Convolutional Neural Network (CNN), and Bidirectional Long Short-Term Memory (Bi-LSTM). The nucleotides were encoded using one-hot encoding, where A is represented as [0,0,0,1], T as [0,0,1,0], G as [0,1,0,0], and C as [1,0,0,0]. The architectures of these three models are detailed in Table 3. The primary distinction among the models lies in their main layer structures. MLP employs six layers of Dense layers with a decreasing number of units through the layers, implementing a dropout rate of 0.3. CNN consists of three Conv1D layers with 100, 100, and 80 filters, respectively, each followed by a MaxPooling1D layer and a dropout of 0.2. BiLSTM employs one layer of Bidirectional LSTM, with 80 units and a dropout of 0.2.

**Table 3.** The architectures of the deep-learning models used for country-based COVID-19 DNA sequences classification.

| Model | Main Layer: Unit/Filter [Dropout] | Fully Connected Layer: Unit, Activation, Dropout | Output Layer | Hyperparameters: Optimizer, Learning Rate, Batch Size |
|---|---|---|---|---|
| MLP | Dense (6 layers): 400, 300, 200, 100, 50, 10 [0.3] | ReLU | Dense (2, softmax) | Adam, 0.001, 1000 |
| CNN | Conv1D (3 layers): 100, 100, 80. MaxPooling1D (3 layers): 4, 2, 2 [0.2]. | 20, ReLU, 0.5 | Dense (2, softmax) | Adam, 0.001, 1000 |
| BiLSTM | BiLSTM (1 layer): 80 [0.2] | 20, ReLU, 0.5 | Dense (2, softmax) | Adam, 0.001, 1000 |

The data are divided into training and testing sets with a split ratio of 90:10. The training set is further divided into 95% for training and 5% for the validation process. This study implemented the model using Keras based on the TensorFlow deep learning library. The loss function was optimized during model training through the Adam optimizer, with a batch size set to 1000 and the epoch set to 100. By applying the early stop strategy, training was stopped if the loss of the validation set did not decrease for 6 epochs. Performance metrics for classification evaluation for each model included accuracy, precision, recall, and F-1 score. For our implementation, we utilized the Kaggle workspace with GPU T4 ×2 accelerators to execute the code. Additionally, we employed the cross-validation technique with a 10-fold and a validation split of 0.2 to calculate classification accuracy for double assurance in evaluating the performance of the MLP model.

## 3. Results and Discussion

Figure 4 displays the accuracy chart of DNA sequence classification for the MLP, CNN, and BiLSTM models. When considering all available sequences, as depicted in Figure 4a, it is shown that sequences from Brunei can be distinguished from those of other ASEAN countries with almost perfect accuracy, reaching 98%. Conversely, the three countries with a lower accuracy are Indonesia, Malaysia, and Vietnam. This pattern remains relatively similar when exclusively involving the Delta variant in the dataset for each country, as shown in Figure 4b. Therefore, even though the variant is the same, the DNA sequences can still be differentiated based on the country of origin.

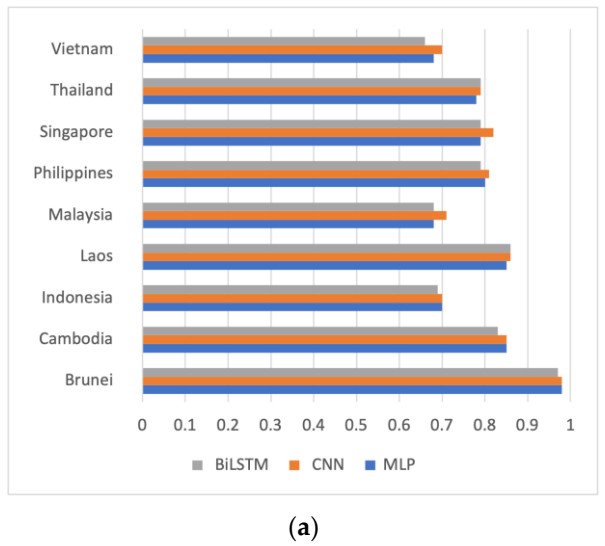
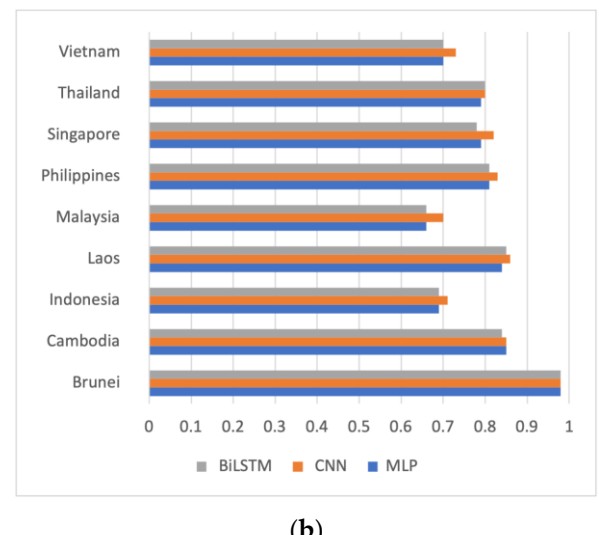

(**a**)      (**b**)

**Figure 4.** Accuracy chart of 'one-vs-all' country-based classification using MLP, CNN, and BiLSTM models (**a**) involving all available sequences and (**b**) only involving Delta variant sequences.

Regarding the impact of different areas among ASEAN countries, while Brunei exhibits a higher sequence accuracy than Malaysia, for instance, it is noteworthy to consider other classification results depicted in Figure 4. In this figure, the sequence accuracy of Singapore, despite being a small country, is not higher than that of Cambodia and Laos, two countries with larger geographical areas. In future work, it would be intriguing to explore the correlation between a country's size and the diversity of its DNA sequences for generalizing this finding.

Table 4 demonstrates the detailed performance metrics for each model, involving Delta variant sequences. The execution time, which measures the time in seconds at which the code is executed for the dataset of each corresponding country, is also calculated. Notably, from Table 4, the execution time of the MLP is relatively faster than that of the CNN or BiLSTM, with a comparable level of accuracy. Hence, the MLP model proves to be suitable in terms of both accuracy and execution time. For further assurance in evaluating the performance of the MLP model, Table 5 shows the classification accuracy resulting from the 10-fold cross-validation technique. It is obvious from Table 5 that the accuracy results are relatively similar to those of the previous scenario where the model was trained without the implementation of cross-validation.

**Table 4.** Execution time and performance metrics for classification evaluation for each model, involving Delta variant sequences.

| Country Name | MLP Model | | | | | CNN Model | | | | | BiLSTM Model | | | | |
|---|---|---|---|---|---|---|---|---|---|---|---|---|---|---|---|
| | Acc. | Prec. | Recall | F1-Score | Time | Acc. | Prec. | Recall | F1-Score | Time | Acc. | Prec. | Recall | F1-Score | Time |
| Brunei | 0.98 | 0.98 | 0.98 | 0.98 | 00:00:58 | 0.98 | 0.98 | 0.98 | 0.98 | 00:02:09 | 0.98 | 0.98 | 0.98 | 0.98 | 0:12:17 |
| Cambodia | 0.85 | 0.87 | 0.85 | 0.85 | 00:01:18 | 0.85 | 0.87 | 0.85 | 0.85 | 00:05:16 | 0.84 | 0.86 | 0.84 | 0.83 | 0:13:29 |
| Indonesia | 0.69 | 0.71 | 0.69 | 0.69 | 00:01:35 | 0.71 | 0.72 | 0.71 | 0.70 | 00:08:54 | 0.69 | 0.7 | 0.69 | 0.69 | 0:23:30 |
| Laos | 0.84 | 0.86 | 0.84 | 0.83 | 00:01:22 | 0.86 | 0.88 | 0.86 | 0.86 | 00:06:14 | 0.85 | 0.88 | 0.85 | 0.85 | 0:10:50 |
| Malaysia | 0.66 | 0.68 | 0.66 | 0.65 | 00:01:26 | 0.70 | 0.72 | 0.7 | 0.69 | 00:11:15 | 0.66 | 0.67 | 0.66 | 0.65 | 0:18:12 |
| Philippines | 0.81 | 0.82 | 0.81 | 0.80 | 00:01:47 | 0.83 | 0.84 | 0.83 | 0.82 | 00:07:58 | 0.81 | 0.83 | 0.81 | 0.81 | 0:20:47 |
| Singapore | 0.79 | 0.79 | 0.79 | 0.79 | 00:01:34 | 0.82 | 0.82 | 0.82 | 0.82 | 00:11:22 | 0.78 | 0.8 | 0.78 | 0.77 | 0:19:29 |
| Thailand | 0.79 | 0.79 | 0.79 | 0.79 | 00:01:40 | 0.80 | 0.80 | 0.80 | 0.80 | 00:07:27 | 0.80 | 0.80 | 0.80 | 0.80 | 0:14:34 |
| Vietnam | 0.70 | 0.71 | 0.70 | 0.70 | 00:01:38 | 0.73 | 0.74 | 0.73 | 0.72 | 00:10:02 | 0.70 | 0.71 | 0.70 | 0.69 | 0:13:30 |

**Table 5.** Execution time and average of the classification accuracy with 10-fold cross-validation for MLP model, involving Delta variant sequences.

| Country Name | Average of Accuracy for All Folds | Standard Deviation | Execution Time |
|---|---|---|---|
| Brunei | 97.97 | 0.05 | 00:16:06 |
| Cambodia | 85.22 | 0.23 | 00:14:07 |
| Indonesia | 66.72 | 0.52 | 00:16:08 |
| Laos | 83.44 | 0.27 | 00:18:33 |
| Malaysia | 65.07 | 0.44 | 00:16:46 |
| Philippines | 78.37 | 0.58 | 00:20:27 |
| Singapore | 78.64 | 0.22 | 00:22:03 |
| Thailand | 78.38 | 0.30 | 00:18:00 |
| Vietnam | 67.01 | 0.31 | 00:21:02 |

We now analyze these classification results in the context of the pandemic situation and the implementation of international travel policy. Initially, we observed the pandemic situation in each country, focusing particularly on the number of cases and deaths accumulated during the period from July to December 2021. Referring to the second dataset from official data collated by Our World in Data [26], as depicted in Figure 5, it appears non-coincidental that Malaysia, one of the countries with a lower accuracy, experienced a severe condition with a notably high number of both accumulated new cases and deaths per million people. In contrast, Cambodia and Laos exhibited relatively lower numbers. While it is premature to conclude that a country with high accuracy, in terms of its DNA sequence, which can be distinguished well from others, tends to undergo a favorable

COVID-19 condition with fewer accumulated cases and deaths, we consider this as trigger information to be explored separately in the future.

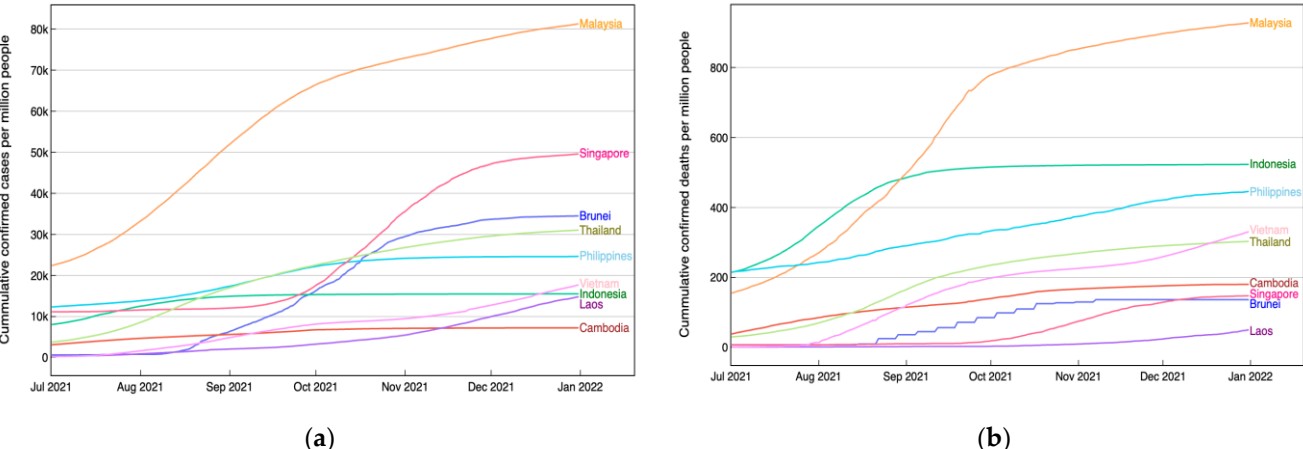

**(a)**                                                                **(b)**

**Figure 5.** (**a**) Cumulative confirmed cases per million people and (**b**) cumulative deaths per million people in ASEAN countries during the second half of 2021.

For additional context, without considering the DNA sequence, a previous study examining pandemic trajectories at the beginning of the outbreak in four ASEAN countries revealed that Malaysia and Vietnam started with strong early performances, while the crisis began severely in the Philippines and Singapore [28]. However, challenges persisted in Malaysia and the Philippines, with relative stability in Vietnam and Singapore. The study emphasized that significant political uncertainty persisted, exemplified by the fragile political situation in Malaysia in 2021. Given that the effectiveness of planning and executing a comprehensive multisectoral response is heavily dependent on national leadership, initial resources obviously play a crucial role, but policy actions also carry significant weight amid the fight against the pandemic.

Moving to the main purpose of the study, we further explored the potential correlation between classification accuracy and policy implementation in each country. We utilized the international travel policy index resulting from the processing of the third dataset, accessed from The Oxford COVID-19 Government Response Tracker (OxCGRT) project [27]. The strict implementation of international travel policy, compared with other government policies, might correlate with the evolution of the virus within a country. Therefore, correlating this policy with classification accuracy provides a preliminary insight into its effectiveness in safeguarding the country from pandemic outbreaks. Before observing the correlation, let us direct our focus to Figure 6, which provides a graphical representation of the monthly international travel policy index in each country throughout 2021.

Taking a closer look at Figure 6, we noticed a consistent application of strict international travel policy by Vietnam throughout 2021, while other countries, particularly Laos, exhibited frequent changes in their restriction scale almost every month. The majority of lines overlap at an index of 75, indicating that most countries implemented bans on arrivals from some regions. Vietnam stands out as the country that implemented a total border closure for more than half of the year, while Singapore noticeably eased restrictions to only screening arrivals in the last four months of 2021. Above all, none of the ASEAN countries had completely open borders during the 2021 timeframe.

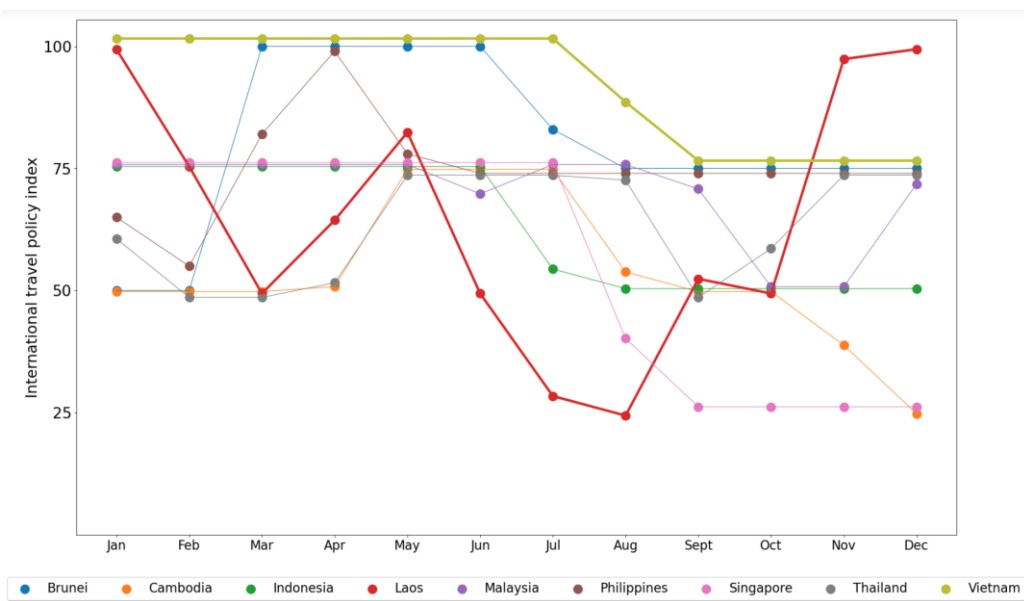

**Figure 6.** Monthly average international travel policy index in each ASEAN country throughout the year 2021.

To investigate whether the international travel policy index correlates with the accuracy of DNA sequence classification within each country, we determined the correlation coefficient between the policy index in a specific month and the classification accuracy in a subsequent month in the second half of 2021. We only considered six countries, excluding Brunei, Laos, and the Philippines due to insufficient monthly datasets. Referring to the emergence of the SARS-CoV-2 Omicron variant [29], where a new wave of infection is expected approximately every additional (up to) 4 months of virus circulation (although we cannot confirm the periodicity will be maintained), we calculated the correlation of the policy index implemented up to 4 months before the collection date of DNA sequences. For an initial observation, Figure 7 illustrates the chart of the monthly policy index implemented up to 4 months before the sequence collection date, alongside the monthly classification accuracy for each country. It is important to note that for this month-wise classification, we conducted a similar classification strategy, but the involved sequence was reduced to 200 sequences for each class with a batch size of 100.

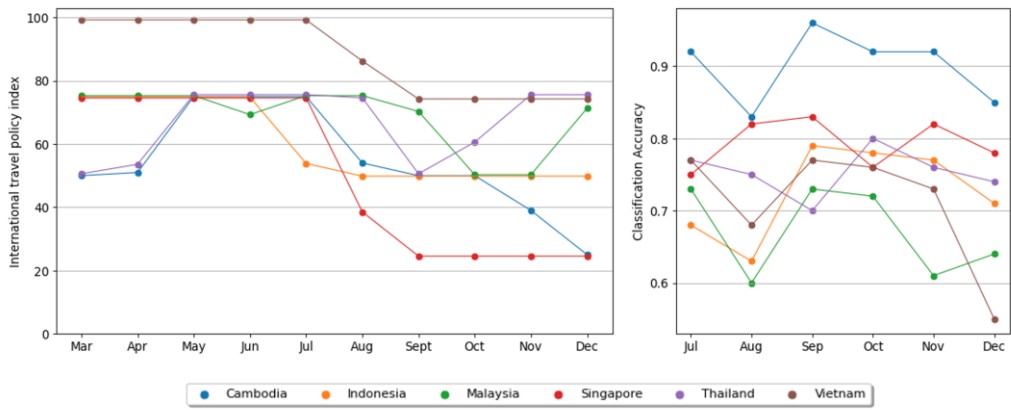

**Figure 7.** The implemented international travel policy index in ASEAN selected countries up to 4 months before the DNA sequence collection date and monthly DNA sequence classification accuracy in corresponding countries.

To indicate a potential link between prior $i$-month policy indices and the accuracy of DNA sequences in the corresponding month, Table 6 shows diverse correlation coefficients for each country. To address the issue of a very small sample size [30], we calculated Pearson correlations between the weekly policy index and the accuracy of week-wise classification, with 25 sequences per class, in a six-months dataset. This calculation means that each entry in Table 6 results from the Pearson correlation and $p$-value calculation involving 24 pairs of data. Except for Indonesia, the majority of correlations demonstrated a positive trend, suggesting that, if a relationship exists, the policy index positively correlates with the classification accuracy.

**Table 6.** Correlation coefficient, $r$, and $p$-value between the week-wise accuracy of DNA sequence classification and the weekly policy index implemented $i$-months prior to the sequence collection date.

| $i$ Months Before | Cambodia | | Indonesia | | Malaysia | | Singapore | | Thailand | | Vietnam | |
|---|---|---|---|---|---|---|---|---|---|---|---|---|
| | $r$ | $p$-Value | $r$ | $p$-Value | $r$ | $p$-Value | $r$ | $p$-Value | $r$ | $p$-Value | $r$ | $p$-Value |
| $i = 4$ | 0.28 | 0.17 | −0.48 | 0.02 | −0.10 | 0.58 | 0.34 | 0.09 | 0.07 | 0.71 | 0.49 | 0.01 |
| $i = 3$ | 0.04 | 0.83 | −0.50 | 0.01 | −0.40 | 0.06 | −0.03 | 0.88 | 0.01 | 0.97 | 0.77 | 0.00 |
| $i = 2$ | 0.08 | 0.68 | −0.50 | 0.01 | 0.32 | 0.12 | 0.26 | 0.21 | −0.34 | 0.09 | 0.20 | 0.34 |
| $i = 1$ | −0.16 | 0.43 | −0.41 | 0.05 | 0.55 | 0.00 | 0.05 | 0.80 | −0.37 | 0.07 | 0.50 | 0.01 |
| $i = 0$ | 0.31 | 0.13 | −0.15 | 0.48 | 0.02 | 0.92 | −0.18 | 0.37 | 0.34 | 0.09 | 0.20 | 0.35 |

Utilizing the $p$-value with a significance level of 5%, we observed a significant relationship between classification accuracy in Vietnam and the policy implemented four months before the collection date of the DNA sequence. Here, the $p$-value equals 0.01, which is less than 5%, indicating that the correlation is statistically significant with a coefficient of 0.49. This suggests that the international travel policy in March positively corelates with classification accuracy in July, while the policy in April corresponds to accuracy in August, and so forth. However, with the same significance level, the correlation between the accuracy and policy index of the preceding 3 months is the highest in Vietnam, standing at 0.77. Therefore, the classification accuracy in Vietnam is highly positively correlated with the policy index three months before the collection date of DNA sequences. Similar to the interpretation for Vietnam, the accuracy in Malaysia exhibits a positive correlation with the policy index one month before the DNA sequence collection date, with a moderate relationship standing at 0.55. Meanwhile, the accuracy in Indonesia is negatively correlated with the policy index up to four months before the DNA sequence collection date.

Unfortunately, at a 5% significance level, it was determined that there is no significant correlation observed between the policy index and DNA sequence classification accuracy for other countries. However, it is crucial to approach the interpretation of data cautiously and not consider a $p$-value of 0.05 as a definitive threshold [31]. Relying on the $p$-value derived from a sole statistical test to assess the scientific validity of research constitutes an inappropriate use of the $p$-value. Nevertheless, it is important to note that alternative approaches to the $p$-value also share comparable limitations [32]. If we are allowed to utilize a significance level set at 15%, similar to Vietnam, DNA sequence classification accuracy in Singapore positively correlates with the policy index applied four months before the sequence collection date. Moreover, classification accuracy in Cambodia and Thailand also positively correlates with its policy implemented in the same month as the collection date, with fair coefficients of 0.31 and 0.34, respectively. While the data show a correlation between the classification accuracy and international travel policy index, it is essential to understand that correlation does not imply causation [30]. Further investigation is required to explore whether the policy is a causative factor for country-based DNA sequence classification accuracy.

Regarding the relationship between the international travel policy index and the accuracy of DNA sequence classification among ASEAN countries, we found it challenging to definitively conclude that a higher policy index in one country correlated with a higher accuracy in distinguishing its sequences from others. This observation is apparent in Figure 7, where Singapore consistently maintains a higher accuracy than Vietnam almost over the entire timeframe, despite having a consistently lower policy index every 4 months preceding sequence collection. In addition, Cambodia, with the highest sequence accuracy throughout the period, does not necessarily exhibit the highest policy index among the focused countries. This suggests the presence of additional influencing factors on the accuracy of DNA sequences between each country. For future research, exploring other aspects that may explain why efforts to boost the international travel policy index in each country may not uniformly correlate with results regarding how DNA sequences significantly differ from those of other countries would be intriguing.

## 4. Conclusions

DNA sequences, even if they belong to the same variant lineage, can be classified based on the country of origin. This study conducted across nine ASEAN countries reveals distinct accuracy levels for each country, indicating the extent to which the sequences can be distinguished from those of other countries. Observing the weekly classification also reveals a moderate correlation between the international travel policy index and the accuracy of DNA sequence classification. Specifically, a higher weekly policy index in a certain prior month correlates with a higher week-wise accuracy within each country. However, when comparing accuracy among countries, it becomes apparent that a deeper understanding of additional factors influencing genetic data is crucial. The variations in classification accuracy between countries do not seem to align with their comparable policy indices. Considering the larger area, which may influence a more diverse character of sequences belonging to a country, the design of this study can be implemented for a group of countries with a similar land area to avoid any bias.

This study is the first attempt to link the genetic landscape of COVID-19 with policies implemented to restrict people's movement, aiming to isolate virus evolution. Although a more scientific approach and detailed debate are needed to determine the level to which COVID-19 has been influenced by policies in different countries, this paper presents significantly meaningful results based on the given evidence. The findings not only contribute to our understanding of COVID-19 but also pave the way for insights into other diseases that have caused serious crises as well as potential future pandemics in ASEAN countries and beyond.

**Author Contributions:** Conceptualization, H.-S.P.; methodology, E.K.; software, E.K.; validation, H.-S.P.; formal analysis, E.K.; data curation, E.K.; writing—original draft preparation, E.K.; writing—review and editing, E.K. and H.-S.P.; visualization, E.K.; supervision, H.-S.P. All authors have read and agreed to the published version of the manuscript.

**Funding:** This research received no external funding.

**Institutional Review Board Statement:** Not applicable.

**Informed Consent Statement:** Not applicable.

**Data Availability Statement:** Publicly available datasets were analyzed in this study. The genome sequences of the SARS-CoV-2 are available from the GISAID database (https://www.gisaid.org/), upon registration. International travel policy dataset can be found here: https://github.com/OxCGRT/covid-policy-dataset/tree/main/data. The number of COVID-19 cases and deaths dataset can be found here: https://github.com/owid/covid-19-data/tree/master/public/data, all accessed on 30 December 2023.

**Acknowledgments:** We would like to express our gratitude to Luqmanul Chakim, Cici Suhaeni and Ika Widiastuti for their insightful discussions during the development of this research.

**Conflicts of Interest:** The authors declare no conflicts of interest.

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
