# Peer review of "Country-Based COVID-19 DNA Sequence Classification in Relation with International Travel Policy"

_applsci, doi:10.3390/app14051916_

Round 1
Reviewer 1 Report
Comments and Suggestions for Authors
Focusing on DNA sequences from nine ASEAN countries, Authors conducted a two-class classification to distinguish sequences from individual countries and mixed sequences from others. The sequences are initially dissected into 200-base pair units, and a deep-learning method is employed to construct a model. The results showcase the capacity to differentiate DNA sequences with varying accuracy for each country. Additionally, the index of policy implemented several months before the sequence collection date in each country moderately correlates with the accuracy value.
As indicated by Authors, this study is the first attempt to link the genetic landscape of COVID-19 with policies implemented to restrict people's movement, aiming to isolate virus evolution. Although a more scientific approach and detailed debate are needed to determine the level to which COVID-19 has been influenced by policies in different countries, this paper presents significantly meaningful results based on the given evidence. The findings not only contribute to our understanding of COVID-19 but also pave the way for insights into other significant diseases in ASEAN countries and beyond.
Please clarify:
"Additionally, it is noteworthy that the execution time of the ANN is relatively faster than that of the CNN or BiLSTM, with a comparable level of accuracy". Is the "execution time" referred to the time of laboratory analysis ?
"The experimental protocol involved a split of 90% for training and 10% for testing".
"Indeed, initial resources play a crucial role, but policy actions also weigh significantly. Therefore, this situation cannot be separated from the fact that Malaysia is unfortunately assessed as not performing well in leadership and governance amid the fight against the COVID-19 pandemic". This quote seems too generic in meaning.
"Employing the p-value method with a significance level of 5%, a significant relationship is discerned between the accuracy in Vietnam and the policy implemented 4 months before the DNA sequence collection date". Why 4 months?
With regard to the last in conclusions:
"The findings not only contribute to our understanding of COVID-19 but also pave the way for insights into other significant diseases in ASEAN countries and beyond". Please indicate specifically "other significant diseases" which will be advantage of that studies.
Comments on the Quality of English LanguagePlease revise and check the English language.
Reviewer 2 Report
Comments and Suggestions for Authors
In the paper “Country-Based COVID-19 DNA Sequence Classification in Relation with International Travel Policy”, Khatizah and Park studied the relationship between COVID-19 DNA sequences and international travel policy. To publish in Applied Sciences, my specific comments are as follows:
1. Table 1: It is not clear whether the reported number of SARS-CoV2 DNA sequences from the nine ASEAN countries represents all the submitted sequences to the GISAID database or only shows the high coverage sequences that the authors have reported to use in their analysis.
2. The authors should add a schematic overflow for the deep learning models exhibiting the data processing, feature encoding, model design, model training, and model testing. It would help to interpret their classification approaches and results better.
3. The authors also did not mention any of the model's hyperparameters. For example, in the case of CNN, what is the size of the kernels or number of kernels used? Similarly, for ANN, the number of nodes or hidden layers, etc. Without all this information, interpreting the models becomes very challenging.
4. The authors reported that they performed “one-vs-all” classification to train their models by randomly selecting 1000 samples from the country of interest and then another 1000 from the rest of the countries. What is unclear from the manuscript here is whether the authors performed this once for each of the countries or for each country, this has been repeated multiple times with different random sample selections as described in the text.
5. For the countries, for which the number of samples is less than 1000, such as Brunei and Laos, what strategy is employed to select 1000 random samples for classification?
6. Line 123: The authors reported that the execution time of ANN is relatively faster than CNN and LSTM. It would be better if the authors could add a plot or table to show that.
7. The authors should have a small diagram or dataset description explaining the policy index data. It is not very clear from the manuscript how they processed the data.
8. For the month-wise classification models, it has not been reported whether the authors used the same classification strategy as before and also how the 1000 random samples were selected for classification.
Reviewer 3 Report
Comments and Suggestions for Authors
The authors of this paper investigate the ability to classify the COVID-19 sequences of each of the nine ASEAN countries to determine if they can be clearly categorized from the rest. In addition, they examine whether this classification accuracy correlates with the policies of each country. While the endeavor is indeed challenging, there are several significant problems with the methodology employed. First, it would be more appropriate in such cases to use cross-validation to calculate classification accuracy. More fundamentally, I believe it is too simplistic to assume a direct correlation between the machine learning model and sequence accuracy. For example, larger countries such as Malaysia would naturally have greater COVID-19 diversity, making classification more complex. In contrast, smaller countries like Brunei may have less diversity due to their limited geography, potentially simplifying the classification task.
Reviewer 4 Report
Comments and Suggestions for Authors
In this manuscript the authors examine the classification of COVID-19 DNA sequences based on their country of origin across nine ASEAN countries. The authors observe distinct accuracy levels for each country in indicating the extent to which the sequences can be distinguished from those of other countries and find a correlation between the international travel policy index and the accuracy of DNA sequence classification. The results presented in the current study are significant and hence the manuscript can be accepted for publication.
Reviewer 5 Report
Comments and Suggestions for Authors
This paper is relatively well written and the research it reports appears well done. Here are a couple of items to attend to in a revision.
Minor: On first use, the label ASEAN should be spelled out in full.
Also, in the Abstract, the statement "Additionally, the index of policy implemented several months before the sequence collection date in each country moderately correlates with the accuracy value." needs to be restated to make more clear what aspects of travel policy specifically are referenced.
Given the importance of the travel policy index to the analysis reported in the paper, it also needs to be given more explicit statements and attention in the paragraph that begins in line 154 and the subsequent paragraphs, including the Conclusions section.
Also, the visual distinctions among the temporal graphs of the 9 countries in Figure 4 are not very clear. If something can be done to improve this, that would be very good.
Round 2
Reviewer 2 Report
Comments and Suggestions for Authors
The authors have successfully answered all of my concerns.
Reviewer 5 Report
Comments and Suggestions for Authors
The revisions to this manuscript have been responsive to the previous review. I have no further suggestions for revision.
Comments on the Quality of English LanguageThe manuscript needs only standard review/edits of the English grammar.